# Image Recognition of Wind Turbine Blade Defects Using Attention-Based MobileNetv1-YOLOv4 and Transfer Learning

**DOI:** 10.3390/s22166009

**Published:** 2022-08-11

**Authors:** Chen Zhang, Tao Yang, Jing Yang

**Affiliations:** 1Hubei Engineering Research Center for Safety Monitoring of New Energy and Power Grid Equipment, Hubei University of Technology, Wuhan 430068, China; 2School of Energy and Power Engineering, Huazhong University of Science and Technology, Wuhan 430074, China; 3School of Energy and Power Engineering, Changchun Institute of Technology, Changchun 130012, China

**Keywords:** wind turbine blades, image recognition, MobileNetv1-YOLOv4, attention-based, transfer learning

## Abstract

Recently, the machine-vision-based blades surface damage detection technique has received great attention for its low cost, easy operation, and lack of a need for prior knowledge. The rapid progress of deep learning has contributed to the promotion of this technology with automatic feature extraction, a broader scope of application, and stronger expansibility. An image recognition method of wind turbine blade defects using attention-based MobileNetv1-YOLOv4 and transfer learning is proposed in this paper. The backbone convolution neural network of YOLOv4 is replaced by the lightweight MobileNetv1 for feature extraction to reduce complexity and computation. Attention-based feature refinement with three distinctive modules, SENet, ECANet, and CBAM, is introduced to realize adaptive feature optimization. To solve the problem of slow network convergence and low detection accuracy caused by insufficient data, a two-stage transfer learning approach is introduced to fine-tune the pre-trained network. Comparative experiments verify the efficacy of the proposed model, with higher detection accuracy but a significantly faster response speed and less computational complexity, compared with other state-of-the-art networks by using images of the wind turbine blades taken by an unmanned aerial vehicle (UAV). A sensitivity study is also conducted to present the effects of different training dataset sizes on the model performance.

## 1. Introduction

Under the pressure of excessive greenhouse gas emissions and global climate deterioration, governments around the world have introduced energy structure transformation strategies to vigorously reduce the proportion of fossil energy in the energy supply. As one of the most promising renewable energy resources, wind power has shown great development potential for its relatively high technological readiness level, abundant availability, and relatively low environmental footprint [1]. Statistics from the Global Wind Energy Council (GWEC) show that by the end of 2021, the cumulative installed capacity of global wind power reached 837 GW with an average annual compound growth rate of 11.6% over the past five years. It is estimated that wind power has helped the world to avoid over 1.2 billion tonnes of CO_2_ annually—equivalent to the annual carbon emissions of South America. Moreover, new installations of 557 GW are forecast for the next five years [2]. As a leader in the global wind power market (holding about 1/3 of the share), China has demonstrated its ambitions in this field. In 2021, China′s new installed capacity was 47.57 GW, nearly four times that of the second-ranked country, the United States, which accounted for more than half (50.91%) of the total global newly installed capacity. In the same year, China’s annual wind power generation was 652.6 GWh, which is a 40.5% year-on-year increase and satisfies approximately 7.5% of its social electricity consumption. It is also predicted that non-fossil energy will account for about 25% of primary energy consumption by 2030. At that time, the cumulative installed capacity of wind power and photovoltaic power generation will reach more than 1200 GW. 

Despite the massive expansion of the wind power industry, higher requirements for wind power equipment reliability, operation and maintenance (O&M) management have also been put forward. Irregular load patterns, intermittent duration, and harsh weather conditions are main reasons for the degradation and failure of wind turbine (WT) components [3]. The O&M cost accounts for approximately 10–15% of the total income for an onshore farm and 20–25% for offshore WTs for 20 years of operating life [4]. As the core components for WTs to capture wind energy, blades play a vital role in the performance of the whole system. The size of WT blades (WTBs) has been increasing in recent years with the development of manufacturing technology (up to 107 m), leading to greater efficiency and energy production, but presenting higher failure probability [5]. Normally, WTBs are exposed to harsh circumstances, being affected by moisture, sand, snow and lightning strikes. Contamination on the blade surface will lead to the corrosion of the coating material and accelerate the weathering and oxidation of the blade surface, resulting in the formation of fine cracks. In snow and ice, WTBs are prone to freezing when the saturation air temperature is below −20 °C, which reduces its strength and toughness, and increases aerodynamic resistance. Wind and sand may cause spot defects, holes, surface peeling, fiber layer damage or cracking, while corrosive salt spray destroys the protective layers of the blade and causes aerodynamic imbalance after combining with other particles. A large number of spots and holes will form a more harmful pitted surface, which enhances conductivity and easily attracts lightning strikes in thunderstorms, or quickly forms a brittle layer in cases of high temperature and strong wind. The further growth of the crack defects will cause the peeling of the surface coating and the spontaneous cracking of internal bonding, which results in the aerodynamic imbalance and overall fracture. A detailed report of wind farm operation revealed that the downtime likelihood for each failure of wind turbine blades can reach 8 to 62 h/year per turbine [6]. In China, the manufacturing cost of the blade accounts for more than 20% of the total cost of the wind turbine [7]. 

To find defects on the blade surface as early as possible, a number of damage detection technologies have been researched and employed, such as strain measurement [8,9,10], acoustic emission [11,12], ultrasound [13,14], vibration [15,16,17] and thermography [18,19]. The above methods have achieved good results in fault feature recognition, experimental verification and small-scale applications. Nevertheless, being susceptible to environmental influences, changing the stiffness of the blade structure, altering boundary conditions, influencing dynamic properties, and requiring expensive sensors and prior expertise are significant drawbacks. With the development of image processing approaches, the detection technology based on machine vision has become a promising technique for its low cost, easy operation and lack of a requirement for prior knowledge [20]. So far, machine-vision-based detection technology has gone through two obvious stages. In the first stage, complex manual feature extraction methods were implemented, e.g., Histogram of Gradient (HOG), Local Binary Pattern (LBP), and Scale Invariant Feature Transform (SIFT). Then, classifiers such as Support Vector Machine (SVM), LogitBoost, and Decision Tree (DT) were used to identify the type of defects. As the convolutional neural network (CNN) shined in the 2012 ImageNet classification competition, machine-vision-based detection technology has officially entered the second stage—deep learning, which models high-level abstractions using multiple layers with nonlinear information processing units. As an end-to-end, black-box hierarchical system, it integrates the two basic elements of image recognition, namely, feature extraction and classification, into an adaptive learning framework [21]. A deep CNN for automatically predicting the stress intensity factor (SIF) at the crack tip through computational vision was designed in [22]. Experimental results verified the method with remarkable results of more than 96% of the predicted SIF values falling within 5% of the true SIF values. A vision-based method using the deep architecture of CNNs for detecting concrete cracks without calculating the defect features was designed in [23]. Comparative studies of the proposed CNN using traditional Canny and Sobel edge detection methods were conducted and the results show its advancement and validity to find concrete cracks in realistic situations. A YOLOv3-based model for the recognition of wind turbine blade damage from surveillance drone images was proposed in [24] to resolve the poor detection accuracy problem associated with conventional methods. Experiments showed that the proposed YOLOv3 solution had the fastest average detection speed of 0.2 s and a maximum average precision of 0.96 over other object-detection techniques. A two-stage approach, crack location and crack contour detection, for precisely detecting wind turbine blade surface cracks via analyzing blade images was discussed in [25]. A method for identifying the surface defects of wind turbine blades based on the Mask-RCNN algorithm was proposed in [26]. The convergence of the model was accelerated through the pre-training of the Coco dataset, and the automatic recognition of defects in the blade image was realized and verified in the images inspected by UAV. A method for wind turbine blade damage classification and detection without using transfer learning but by training the model from the image dataset prepared by image augmentation methods and manual editing was discussed in [27]. An accuracy of around 94.94% for binary fault classification and 91% for multiple class fault classification were achieved. A new version of the local binary pattern, called completed local quartet patterns, was proposed to extract fabric image local texture features in [28]. The proposed approach had high accuracy, at or above 97.2 percent, in patterned fabrics with different pattern types.

At present, UAVs equipped with digital cameras have been widely used in wind farms to replace fixed high-definition cameras and manual handheld photography, which can be remotely controlled to freely capture the surface of blades and wirelessly transmit the captured images or videos. Despite the relatively high level of accuracy achieved by the above-mentioned image processing models, the large computational complexity and low detection response speed have become great obstacles hindering the further development of this technology. RCNN series algorithms [26,29] based on region proposal networks and YOLO/SSD/RetinaNet [30,31,32] based on regression have general issues of low real-time capability, massive computation requirements, occupying a large amount of hard disk space and needing high-performance GPU support during implementation. Moreover, the lack of publicly available wind turbine blade damage datasets makes it impossible to train a highly accurate and robust model.

To address the above issues, an image recognition method for wind turbine blade defects using attention-based MobileNetv1-YOLOv4 and transfer learning is proposed in this paper. The backbone convolution neural network of YOLOv4 is replaced by the lightweight MobileNetv1 for feature extraction to reduce complexity and computation. Attention-based feature refinement with three distinctive modules, SENet, ECANet, and CBAM, is introduced to realize adaptive feature optimization. To solve the problem of slow network convergence and low detection accuracy caused by insufficient data, a two-stage transfer learning technique is introduced to fine-tune the pre-trained network. Comparative experiments verify the efficacy of the proposed model with higher detection accuracy but significantly faster response speed and less computational complexity, compared with other state-of-the-art networks, by using images of the wind turbine blades. A sensitivity study is also conducted to present the effects of different training dataset sizes on the model performance.

## 2. Description of Methodology 

The framework of wind turbine blade defect detection using attention-based MobileNetv1-YOLOv4 and transfer learning is presented in Figure 1. The image sets are divided into two parts, the training set and test set, after filtering, labeling, augmenting and preprocessing. Several data augmentation methods (e.g., crop, flips, and distortion) are implemented. The attention-based MobileNetv1-YOLOv4 is first pre-trained by labeled images from the PASCAL VOC dataset, then a two-stage transfer learning technique is introduced to fine-tune the parameters of the network by the training set. The test set is used to evaluate the performance of the trained model, and the prediction results (the defect class and accurate location) are finally obtained. 

### 2.1. Image Preparation and Augmentation 

The original images of wind turbine blades presented in this paper were taken from a 1.5 MW onshore wind farm located in Guangdong province along the southeast coast of China by the Mavic Air 2 UAV with a resolution of 4032 × 3024 and maximum endurance of 34 min. To ensure the clarity of images and safety of the drone, the shooting distance was selected as 8–10 m. A total of 122 images with surface defects were preserved after eliminating the useless information, which were generally classified into four classes: surface spalling (10 images), pitting (17 images), crack (60 images) and contamination (35 images). It is quite clear that these datasets are far from enough to train an accurate and robust deep learning model. Taking into account the fact that great uncertainties exist in the actual shooting, fifteen kinds of augmentation operation, such as adding pixels and noise, flipping, distortion, scaling, changing contrast, and brightness, are introduced to simulate different shooting angles and light conditions as much as possible [33]. An example of data augmentation for a crack defect is shown in Figure 2, for which 15 data enhancement methods are adopted.

### 2.2. Attention-Based MobileNetv1-YOLOv4 

#### 2.2.1. MobileNetv1-YOLOv4 Network

As part of the one-stage end-to-end detection frameworks, the YOLO family (first put forward by Redmon and Farhadi [34]) is a popular series of approaches for object detection, which treat the object detection as a regression problem and predict bounding boxes and class probabilities in a full image [35]. Proposed in April 2020, YOLOv4 perfectly integrates the detection speed and accuracy by referencing the network structure of CSPNet and introducing the Mish activation function, Path Aggregation Network (PANet) and Spatial Pyramid Pooling (SPP). The YOLOv4 network has been used in a variety of scenarios (e.g., the identification and detection of traffic signs, pedestrians and vehicles, and industrial parts), and achieved good results. However, when the traditional convolutional network model runs on embedded devices, problems such as insufficient device memory, slow response, or even unavailability will occur.

As a lightweight and efficient convolutional neural network proposed by Google, MobileNetv1 uses the depthwise separable convolution to split standard convolution into depthwise convolution and pointwise convolution, as shown in Figure 3. N convolution kernels (with a scale of $D_{k}\times D_{k}$ and channel number of M) can be decomposed into M convolution kernels (with a scale of $D_{k}\times D_{k}$ and channel number of 1) and N convolution kernels (with a scale of 1×1 and channel number of M).

The computation cost of this operation by MobileNetv1 is: (1)$$D_{k}\cdot D_{k}\cdot M\cdot D_{F}\cdot D_{F}+M\cdot N\cdot D_{F}\cdot D_{F}$$

By contrast, the cost for convolutional neural network is:(2)$$D_{k}\cdot D_{k}\cdot M\cdot N\cdot D_{F}\cdot D_{F}$$
where M is the number of input channels, N is the number of output channels, $D_{k}$ is the kernel size of a convolution operation, and $D_{F}$ is the feature map size. 

A reduction in computation can be obtained by combining (1) and (2) as:(3)$$\frac{D_{k}\cdot D_{k}\cdot M\cdot D_{F}\cdot D_{F}+M\cdot N\cdot D_{F}\cdot D_{F}}{D_{k}\cdot D_{k}\cdot M\cdot N\cdot D_{F}\cdot D_{F}}=\frac{1}{N}+\frac{1}{D_{k}{}^{2}}$$

Under normal circumstances, the number of output channels N is very large. Therefore, 8–9 times lower computation costs than a standard convolution can be saved for a 3 × 3 depthwise separable convolution.

After each convolution, Batch Normalization (BN) and ReLU blocks are applied; the block structure of depthwise separable convolution can be seen in Figure 4 [35].

The model structure of MobileNetv1-YOLOv4 is illustrated in Figure 5. Compared with the conventional YOLOv4, the feature extraction network (called as Backbone) is replaced by MobileNetv1 to reduce the number of parameters in the network structure. Continuous downsampling is implemented five times on the input image (416 × 416) to obtain a set of feature layers (208 × 208, 104 × 104, 52 × 52, 26 × 26, and 13 × 13). The Mish activation function is used to speed up the training process and achieve a strong regularization effect. The feature fusion network (called Neck) consists of two parts, PANet and SPP. On one hand, the output of the Backbone network is convolved three times as the input of the SPP, which performs maximum pooling and fusion operations of different scales on the input and significantly increases the receptive field, separates out the most significant context features, and causes almost no reduction in the network operation speed. On the other hand, the output of SPP is upsampled twice, merged with the fourth and third feature layers, respectively, and five convolution operations are performed to obtain the feature layers N1 and N2. The N2 layer is then downsampled twice, merged with the output of the N1 layer and SPP, and five convolution operations are performed. To further achieve a light weight, the standard convolutions in Neck are all replaced by depthwise separable convolutions. Three YOLO Heads at different scales (52 × 52, 26 × 26, and 13 × 13) are finally obtained to detect objects of different sizes. Each YOLO Head contains three priori boxes. The prediction layer first predicts the three feature layers output by PANet, then analyzes the target information in each priori box, and finally uses the method of non-maximum suppression to determine the final prediction box. 

The loss function for MobileNetv1-YOLOv4 consists of three parts, confidence loss $L_{Confidence}$, classification loss $L_{Class}$ and bounding box location loss $L_{CIoU}$:(4)$$Loss=L_{Confidence}+L_{Class}+L_{CIoU}$$

$L_{Confidence}$ signifies the confidence error:(5)$$L_{Confidence}=-\sum_{i=0}^{S^{2}}\sum_{j=0}^{B}I_{ij}^{obj}[{\bar{C}}_{i}^{j}\log(C_{i}^{j})+(1-{\bar{C}}_{i}^{j})\log(1-C_{i}^{j})]-\lambda_{noobj}\sum_{i=0}^{S^{2}}\sum_{j=0}^{B}I_{ij}^{noobj}[{\bar{C}}_{i}^{j}\log(C_{i}^{j})+(1-{\bar{C}}_{i}^{j})\log(1-C_{i}^{j})]$$
where $S^{2}$ is the number of grids in the input image, B is the number of bounding boxes generated by each grid, $I_{ij}^{obj}$ and $I_{ij}^{noobj}$ indicate whether the object falls into the bounding box or not, ${\bar{C}}_{i}^{j}$ is the true value of confidence while $C_{i}^{j}$ is the predicted value, and $\lambda_{noobj}$ is the weight coefficient.

$L_{Class}$ refers to the classification error, which is expressed as follows:(6)$$L_{Class}=-\sum_{i=0}^{S^{2}}I_{ij}^{obj}\sum_{c\in classes}\left[{\bar{P}}_{i}^{j}(c)\log(P_{i}^{j}(c))+(1-{\bar{P}}_{i}^{j}(c))\log(1-P_{i}^{j}(c))\right]$$
where ${\bar{P}}_{i}^{j}(c)$ is the true probability that the object belongs to the class, while $P_{i}^{j}(c)$ is the predicted probability.

One improvement of YOLOv4 over its previous version is the introduction of a novel loss function, CIoU, with a faster convergence speed and better performance on bounding box regression, which imposes the consistency of aspect ratios for bounding boxes [36]. The bounding box location loss $L_{CIoU}$ can be expressed as:(7)$$L_{CIoU}=1-IoU+\frac{\rho^{2}(b,b^{gt})}{c^{2}}+\alpha \nu$$
(8)$$IoU=\frac{\left|B\cap B^{gt}\right|}{\left|B\cup B^{gt}\right|}$$
(9)$$\alpha =\frac{\nu }{(1-IoU)+\nu }$$
(10)$$\nu =\frac{4}{\pi^{2}}{\left(\arctan\frac{w^{gt}}{h^{gt}}-\arctan\frac{w}{h}\right)}^{2}$$
where B=(x,y,w,h) and $B^{gt}=(x^{gt},y^{gt},w^{gt},h^{gt})$ represent the centroid coordinate, width, and height of the prediction bounding box and ground truth, respectively, IoU is the intersection over union of the prediction box and ground truth box, b and $b^{gt}$ are the centroid of B and $B^{gt}$, c is the diagonal length of the smallest enclosing box covering ground truth and prediction bounding boxes, α is a positive trade-off parameter, and ν computes the consistency of the aspect ratio.

#### 2.2.2. Attention-Based Modules

As a general auxiliary module of deep learning models, the attention mechanism has been widely used in tasks such as object detection, object tracking and re-identification, which assigns different weights to each piece of information to reflect the importance and relevance of different features. Common attention mechanisms include the channel attention mechanism, spatial attention mechanism, and mixed domain attention mechanism. Three attention-based modules, namely, SENet, ECANet, and CBAM, are discussed in this paper.SENet

SENet (Squeeze-and-Excitation Network) was first proposed in 2017 to improve the quality of representations produced by a network by explicitly modeling the interdependencies between the channels of its convolutional features [37]. It has a simple structure and certain flexibility, and can be embedded into existing networks, expanding the perception range of feature maps to global information. Three sub-processes, Squeeze, Excitation, and Reweight, are sequentially implemented after the convolutional operation $F_{tr}$, as shown in Figure 6. $F_{sq}(\cdot )$ is a squeeze function that performs average pooling on an individual channel of feature map U and produces a $1\times 1\times C_{2}$ dimensional channel descriptor. An excitation function ($F_{ex}(\cdot ,w)$) is a self-gating mechanism made up of three layers, two fully connected layers and a Sigmoid non-linearity layer, after which weights corresponding to each channel are obtained. By applying the excited output on the feature map U, U is scaled ($F_{sq}(\cdot )$) to generate the final output ($\tilde{x}$) of SENet [38].

ECANet
As another implementation of the channel attention mechanism, the ECANet (Efficient Channel Attention Network) is regarded as an improved version of the SENet. It generates channel attention through a fast 1D convolution, the kernel size of which can be adaptively determined by a non-linear mapping of channel dimensions, as shown in Figure 7. Channel-wise global average pooling (GAP) is first implemented on the input feature without dimensionality reduction. Local cross-channel interaction is then captured by fast 1D convolution of size k. The kernel size k represents the coverage of local cross-channel interaction, which is adaptively determined and proportional to the channel dimensions [39]. Then, the Sigmoid function is used to generate the weights of each channel, and the final output is obtained by combining the input feature with channel weights (the same as the final process in the SENet).

CBAM
The CBAM (Convolutional Block Attention Module) is a simple yet effective attention module for feed-forward convolutional neural networks. The attention map of the feature map from two separate dimensions, channel and spatial, is calculated and then multiplied to the feature map for adaptive learning. The channel attention module compresses the spatial dimension of the input feature map, gathers spatial information, retains background and texture information. As shown in Figure 8, the input feature is operated by average-pooling and max-pooling, respectively, and then forwarded to a shared multi-layer perceptron. The channel attention feature map $M_{c}$ is finally generated after merging the two obtained output feature vectors using element-wise summation. The channel attention is computed as:(11)$$M_{c}(F)=\sigma (MLP(AvgPool(F))+MLP(MaxPool(F)))=\sigma (W_{1}(W_{0}(F_{\mathrm{avg}}^{c}))+W_{1}(W_{0}(F_{\max}^{c})))$$
where σ is the Sigmoid function, MLP is the multi-layer perceptron, and $W_{0}$ and $W_{1}$ are the weight matrix.

By contrast, the spatial attention module (shown in Figure 9) uses the spatial relationship of the features and focuses on the information in the two-dimensional input map, which tends to pay more attention to the position information of the input feature map. The average-pooling and max-pooling operations along the channel axis are applied to the refined features from the channel module, which are then concatenated and convolved by a standard convolution layer. The spatial attention module is generated through a Sigmoid function. The spatial attention is computed as:(12)$$M_{s}(F)=\sigma (f^{7\times 7}([AvgPool(F);MaxPool(F)]))=\sigma (f^{7\times 7}([F_{\mathrm{avg}}^{s};F_{\max}^{s}]))$$
where $f^{7\times 7}$ represents a convolution operation with the filter size of 7 × 7.

After the two stages of processing described above, the overall attention process can be summarized as:(13)$$F^{C}=M_{C}(F)⊗F;F^{CS}=M_{S}(F^{C})⊗F^{C}$$
where denotes element-wise multiplication, $F^{C}$ is the channel-refined feature, and $F^{CS}$ is the final refined feature.

#### 2.2.3. Attention-Based MobileNetv1-YOLOv4

In the YOLOv4 network discussed above, the feature extraction network is replaced by MobileNetv1. As a plug-and-play tool, the attention-based module can be flexibly used in the deep neural network to increase the weights of useful features while suppressing the weights of invalid features, paying more attention to target regions containing important information, suppressing irrelevant information, and improving the overall accuracy of target detection. This paper separately adds the attention-based module to each of the three branches at the input of the feature fusion network (red section), in the middle of the feature fusion network (green section), and at the output of the feature fusion network (blue section). The improved network structure is shown in Figure 10. 

#### 2.2.4. Transfer Learning

Transfer learning refers to the application of knowledge/experience learned in one field to a new field to solve new problems. In the field of image classification, transfer learning is a technique that updates and modifies the parameters (e.g., weights and bias) of a model trained on a specific dataset to adapt it to the target dataset to solve other classification problems. The image features are extracted layer by layer through convolution operations in the training process of the neural networks, such that shallow layers are mainly used to extract general feature information, while specific feature information is extracted by deep layers with better composite effects. Therefore, by fine-tuning the parameters of the deep layers of pre-trained models, the transfer learning method is recognized to speed up the training process of the network model and improve the network performance, especially suitable for situations with an insufficient training dataset [40].

The proposed network was pre-trained on a large related dataset—the PASCAL VOC dataset—and then a two-stage transfer learning technique was introduced. In the first stage, the transfer of the Backbone layer is frozen and augmented blade images obtained above are used to tune the hyper-parameters of the Neck layer and YOLO Head. Although the images of the blades with surface defects are quite different from those of the PASCAL VOC dataset, the previous Backbone structures trained in the ImageNet dataset can still extract the local features (such as edge, color, speckle, or textural features) of blade defects [35]. Then, in the second stage, shallow layers and deep layers are combined for coordination training, such that the parameters of the entire network are fine-tuned by the training set.

## 3. Results

In this section, specific examples from a wind farm located along the southeast coast of China are presented to evaluate the performance of the proposed model in the image recognition of wind turbine blade defects. A pre-trained model based on the PASCAL VOC dataset is first loaded. Then, a total of 1952 blade images acquired by the data augmentation methods presented above with four categories of defects (shown in Figure 11) are used to train the model. The dataset is divided into the training set, validation set, and test set according to the ratio of 5:2:3. The hardware environment, software version and training network parameters for this experiment are shown in Table 1. The Early Stopping mechanism is also adopted to avoid over-fitting, such that the training process is shut down once the decreasing value of two adjacent validation losses is lower than 10^−4^ or there is no improvement in validation losses for five consecutive epochs.

Variations of training loss and validation loss over epochs during the training process for MobileNetv1-YOLOv4 are demonstrated in Figure 12. Thanks to the loading of a pre-trained model, the loss value can be reduced to a lower value in a few iterations. As the number of epochs increases, the training loss and validation loss, respectively, converge close to a steady value. The Early Stopping is triggered when the numbers of epochs reaches 73 and the training loss and validation loss finally drop to 2.804 and 2.463, respectively. It is quite clear that the overall training effect is ideal. Meanwhile, the qualitative results by MobileNetv1-YOLOv4 are shown in Figure 13, demonstrating that the proposed model shows good target detection performance for multiple defects.

To measure the performance of different models, the parameter size, FPS (frames per second), precision, recall, $F_{1}$ score, and mAP are introduced as evaluation metrics, given by (14)–(18). The parameter size represents the number of model parameters to be trained. FPS indicates how many images the model can process in one second, which is an important measure for industrial detection. Precision refers to the ratio of the number of objects correctly detected to the number of total objects detected, and recall is the ratio of the number of objects correctly detected to the number of ground truth objects. Generally, precision and recall are two parameters with completely opposite trends; a low false negative value leads to a high precision value but a low recall value, while a high false positive value causes a low precision value but a high recall value. The $F_{1}$ score considers both precision and recall and conveys the balance between precision and recall, such that a higher score indicates a better trained model. The PR (precision–recall) curve describes the trade-off between the precision and recall, and AP (average precision) is the area under the precision–recall curve, such that a larger area means a higher AP and a better model. Furthermore, the mAP is the mean average precision of all classes specified in the test model. In this paper, the IoU threshold is set as 0.5.
(14)$$Precision=\frac{TP}{TP+FP}$$
(15)$$Recall=\frac{TP}{TP+FN}$$
(16)$$F_{1}=2\times \frac{Precision\times Recall}{Precision+Recall}$$
(17)$$AP=\int_{0}^{1}p(r)dr$$
(18)$$mAP=\frac{1}{N}\sum AP_{i}$$
where TP (true positives) is the number of cases that are correctly labeled as positive, TN (true negative) is the number of cases that are correctly labeled as negative, FP (false positive) is the number of cases that are incorrectly labeled as positive, FN (false negative) is the number of cases that are positive but are labeled as negative, and N is the total number of classes.

The classification $F_{1}$ curve and AP curve of these four categories under different score thresholds (IoU) for MobileNetv1-YOLOv4 are presented in Figure 14 and Figure 15. The title indicates the specific figure with a score threshold of 0.5. Finally, the $F_{1}$ score and mAP are obtained at 0.665 and 88.61% for the total defects.

Furthermore, the obtained results for the detection of blade defects are compared with previous works to validate the performance of the proposed model. The model comparison of different detection algorithms is listed in Table 2. Compared with traditional object detection algorithms, the advantages of MobileNetv1-YOLOv4 in terms of model complexity and detection speed are obvious when using the depthwise separable convolution to split the standard convolution. In particular, the parameter size of MobileNetv1-YOLOv4 is reduced to 1/5 that of YOLOv4. In the meantime, the model performances of MobileNetv1-YOLOv4 are even better than those of the FasterR-CNN and YOLOv3 models, with a huge improvement in mAP. Despite the huge reduction in parameter size and training time by YOLOv4-tiny, the poor performance (with an $F_{1}$ score of 0.418 and a mAP of 67.43%) shows that it is simply not appropriate for this detection scenario. However, the simplified operation of convolution by MobileNetv1-YOLOv4 inevitably has a negative impact on the detection performance of the model. Thus, the $F_{1}$ score and mAP of MobileNetv1-YOLOv4 are less than that of YOLOv4 (with a reduction of 0.23 and 3.71%, respectively).

Attention-based feature refinement with three distinctive modules, SENet, ECANet, and CBAM, is introduced in this paper to realize adaptive feature optimization. Three different methods of model integration are discussed and shown in Figure 10. Comparative experiments are conducted with the same dataset and global random seeds, and the results can be seen in Table 3. The addition of the attention-based modules will undoubtedly make the model more complex, increasing the number of parameters to be trained and decreasing the number of detection frames per second. However, as the attention-based module is a lightweight and plug-and-play tool, these negative effects on model complexity are negligible. On the other hand, by assigning different weights to different features, attention-based models achieve better performance over the basic model (MobileNetv1-YOLOv4) with greater precision, recall, $F_{1}$ score and mAP for almost all structures. Among them, the green section (to add the attention-based module in the middle of the feature fusion network) is the optimal model structure with the highest average $F_{1}$ score of 0.811 and mAP of 93.44%, which, respectively, show an increase of 0.146 and 4.83% over the basic model. Moreover, the SENet (with an average $F_{1}$ score of 0.764 and mAP of 92.94) and ECANet (with an average $F_{1}$ score of 0.785 and mAP of 91.90) seem to work better than the CBAM (with an average $F_{1}$ score of 0.745 and mAP of 90.29) for these three model structures. This is mainly because the background and defects of blade training datasets used in the paper are relatively simple, and the channel attention mechanism for focusing on the point feature, line feature, and chiaroscuro feature is adequate and useful. By contrast, the model structure of CBAM is more complex, and learning unnecessary features leads to the degradation of model performance.

The impact of different sizes of training dataset on the performance of detection models is also discussed. Four models, the basic model (MobileNetv1-YOLOv4) and three kinds of attention-based modules in the middle of the feature fusion network, are compared and the results can be seen in Figure 16. It is quite clear that larger datasets correspond to better model performance, that is, a greater $F_{1}$ score and mAP. With the increase in dataset size, the performance of each model gradually stabilizes. Thanks to the Early Stopping mechanism, no over-fitting is found with increasing training dataset size, as shown in Figure 16. Meanwhile, for all training dataset sizes, the $F_{1}$ score and mAP of SENet, ECANet, and CBAM are higher than those of the basic model, and this advantage is more apparent for smaller training sets. Hence, in scenarios where training datasets are insufficient, the attention-based feature refinement may be significantly useful.

## 4. Conclusions and Future Work

An image recognition method for wind turbine blade defects using attention-based MobileNetv1-YOLOv4 and transfer learning is proposed in this paper. The backbone convolution neural network of YOLOv4 is replaced by the lightweight MobileNetv1 for feature extraction in order to reduce complexity and computation. To solve the problems of slow network convergence and low detection accuracy caused by insufficient data, a two-stage transfer learning technique is introduced to fine-tune the pre-trained network. Compared with traditional object detection algorithms, the advantages of MobileNetv1-YOLOv4 in terms of model complexity and detection speed are obvious when using the depthwise separable convolution to split the standard convolution. The mAP of MobileNetv1-YOLOv4 is even greater than that of FasterR-CNN and YOLOv3. However, this simplified operation of convolution inevitably has a negative impact on the detection performance of the model. Although the parameter size of MobileNetv1-YOLOv4 is reduced to 1/5 that of YOLOv4, the $F_{1}$ score and mAP of MobileNetv1-YOLOv4 show a reduction of 0.23 and 3.71%, respectively, compared with that of YOLOv4.

Attention-based feature refinement with three distinctive modules, SENet, ECANet, and CBAM, is introduced to realize adaptive feature optimization. Three different methods of model integration (to add the attention-based module at the input of the feature fusion network, in the middle of the feature fusion network, and at the output of the feature fusion network) are discussed. As a lightweight and plug-and-play tool, the attention-based module reveals negligible negative effects on model complexity, but achieves better performance than the MobileNetv1-YOLOv4, with greater precision, recall, $F_{1}$ score, and mAP for almost all structures. The structure of adding the attention-based module in the middle of the feature fusion network is the optimal model structure with the highest average $F_{1}$ score of 0.811 and mAP of 93.44%, and the SENet and ECANet seem to work better than the CBAM for these three model structures. The impact of different sizes of training dataset on the performance of detection models is also discussed. Four models, the MobileNetv1-YOLOv4 and three kinds of attention-based modules in the middle of the feature fusion network, are compared and results for all training dataset sizes, the $F_{1}$ score, and mAP of SENet, ECANet, and CBAM are higher than those for the MobileNetv1-YOLOv4. It is interesting to find that this advantage is more apparent for smaller training sets, which further verifies the validity and advancement of the attention-based feature refinement.

The proposed deep network can also be used in some other visual defect detection problems, such as lytic bacteriophage detection in sewage water images [43], which can greatly reduce computational complexity while ensuring high detection accuracy. However, despite the valuable findings in the above fields, more complex and full datasets are needed to verify the wide applicability of the above methods. Moreover, the integration of attention-based modules with other detection algorithms also needs be explored further in the future, together with the edge detection method, in order to directly provide detection results using a UAV.

## Figures and Tables

**Figure 1 sensors-22-06009-f001:**
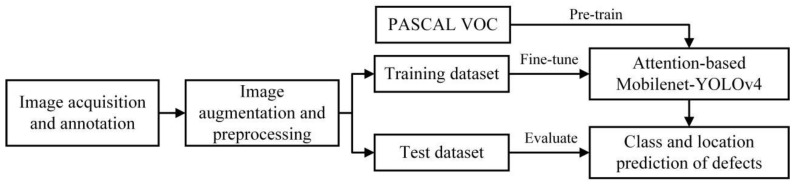
Framework of blade defect detection.

**Figure 2 sensors-22-06009-f002:**
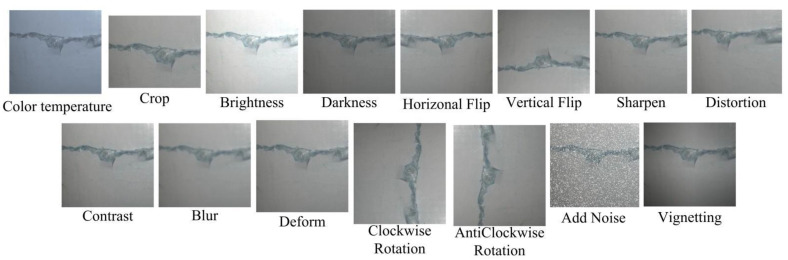
Image augmentation.

**Figure 3 sensors-22-06009-f003:**
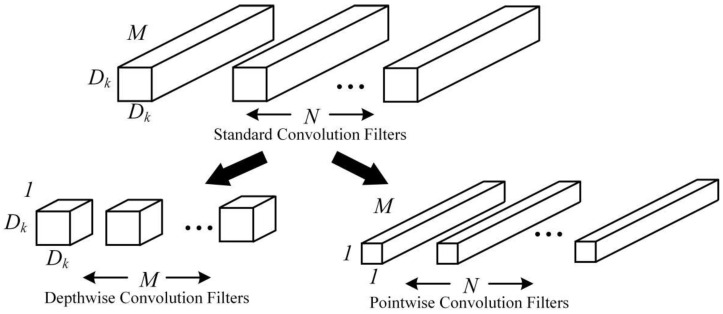
Depthwise separable convolution.

**Figure 4 sensors-22-06009-f004:**
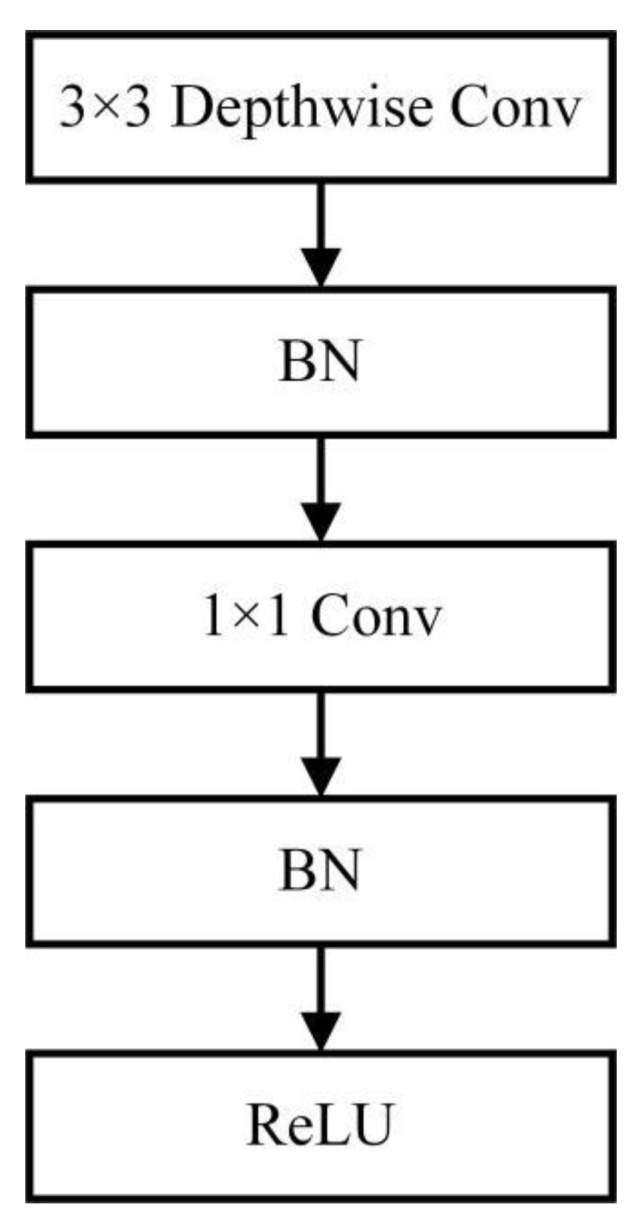
Depthwise separable convolution block structure.

**Figure 5 sensors-22-06009-f005:**
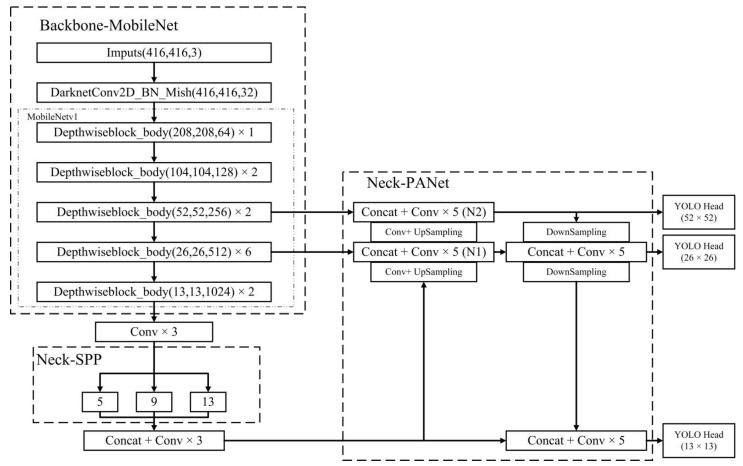
MobileNetv1-YOLOv4 model structure.

**Figure 6 sensors-22-06009-f006:**
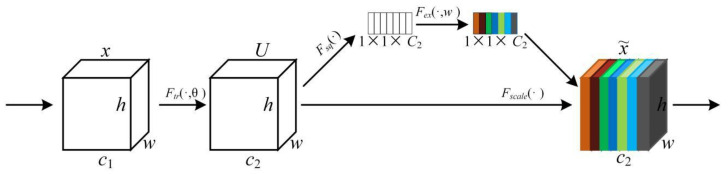
SENet module.

**Figure 7 sensors-22-06009-f007:**
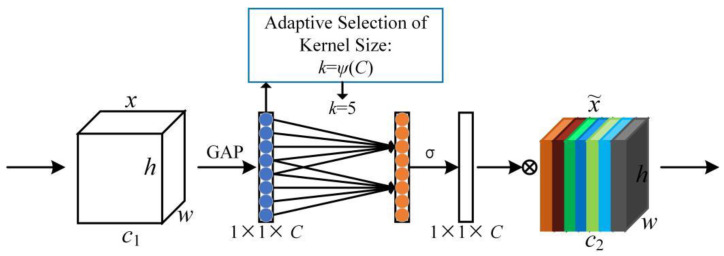
ECANet module.

**Figure 8 sensors-22-06009-f008:**
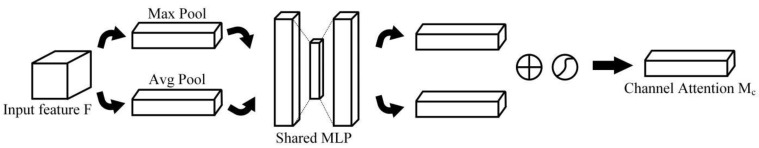
Channel attention module.

**Figure 9 sensors-22-06009-f009:**
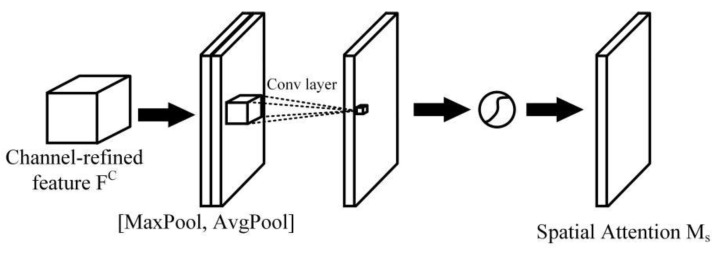
Spatial attention module.

**Figure 10 sensors-22-06009-f010:**
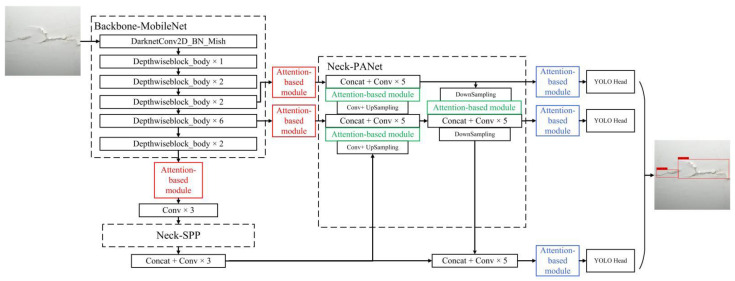
MobileNetv1-YOLOv4 network structure combined with attention-based modules.

**Figure 11 sensors-22-06009-f011:**
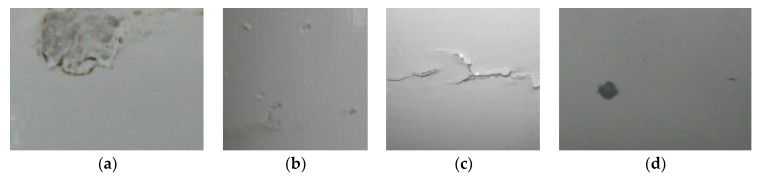
Defect types. (**a**) Spalling. (**b**) Pitting. (**c**) Crack. (**d**) Contamination.

**Figure 12 sensors-22-06009-f012:**
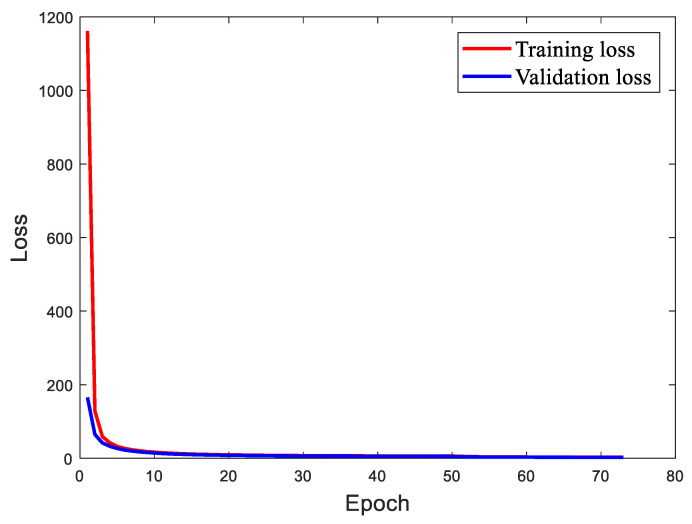
Training and validation loss for MobileNetv1-YOLOv4.

**Figure 13 sensors-22-06009-f013:**
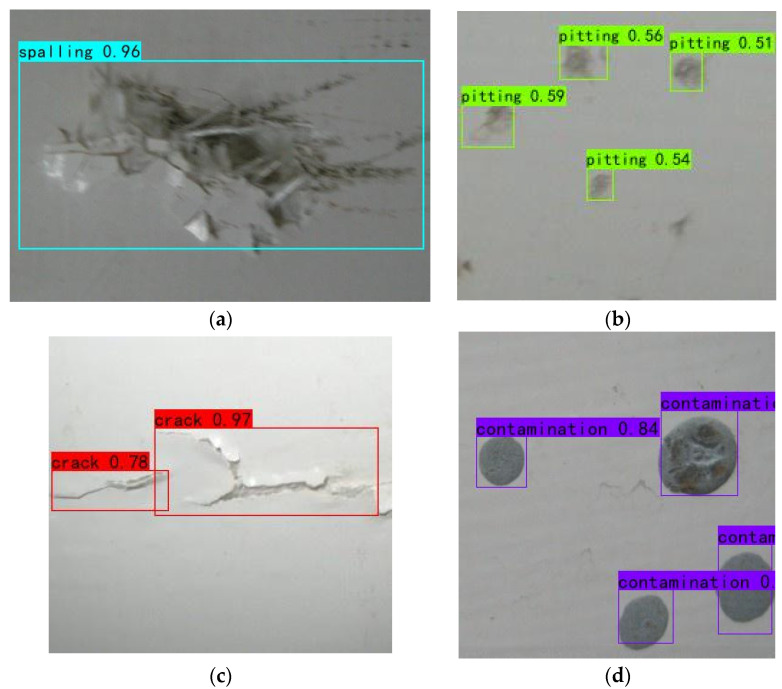
Detection results. (**a**) Spalling. (**b**) Pitting. (**c**) Crack. (**d**) Contamination.

**Figure 14 sensors-22-06009-f014:**
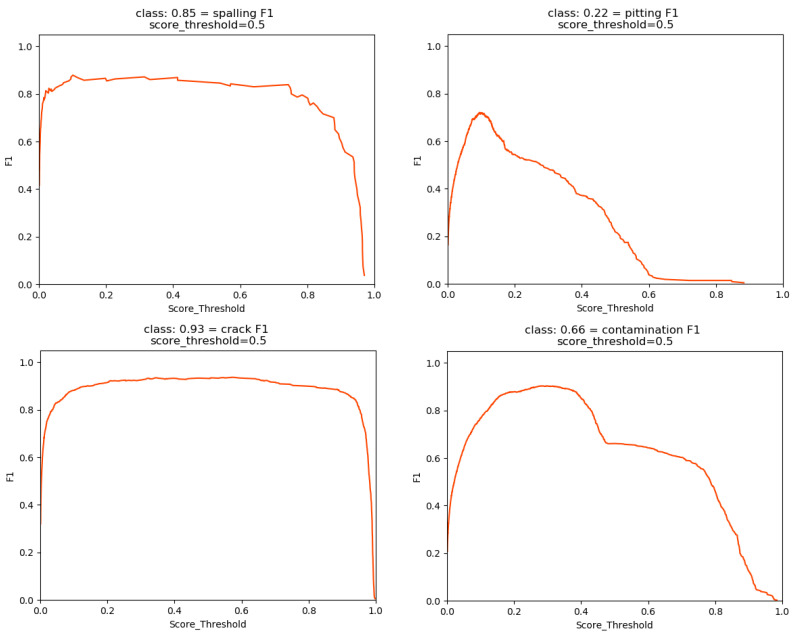
*F*_1_ curve.

**Figure 15 sensors-22-06009-f015:**
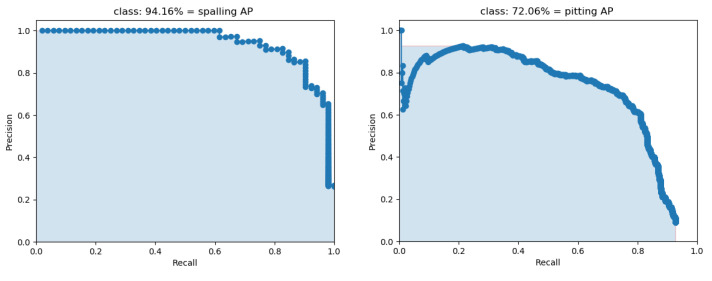
*AP* curve.

**Figure 16 sensors-22-06009-f016:**
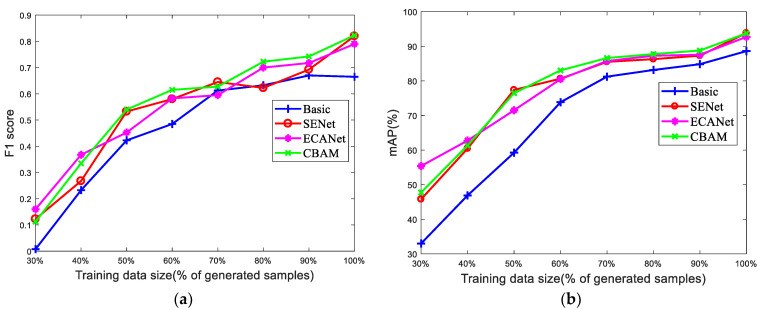
Results for different sizes of training data. (**a**) *F*_1_ score. (**b**) mAP.

**Table 1 sensors-22-06009-t001:** Hardware environment, software version, and training network parameters.

	Configuration
Hardware environment	Operating System: Windows 10
	CPU: Inter E5-2650 v4
	RAM: 64GB
	GPU: NVIDIA 1080Ti
Software version	Pycharm2021.3.2 + Python 3.7.8 + CUDNN11.1.0 + CUDA11.1.74
Training network parameters	Fixed image size: 416 × 416
	Batch size: 16
	Optimizer: Adam
	Learning rate: 10^−3^
	Decay rate: 5 × 10^−^^4^
	Frozen epoch: 50
	Unfrozen epoch: 50

**Table 2 sensors-22-06009-t002:** Model comparison of different algorithms.

Algorithm	Parameter Size (MB)	Training Epochs	Training Time (s)	FPS	Precision (%)	Recall (%)	*F*_1_ Score	mAP (%)
FasterR-CNN [41]	522.91	70	14,060	9.76	69.72	80.80	0.740	81.72
YOLOv3 [24]	236.32	73	7223	19.38	85.93	64.61	0.733	81.81
YOLOv4 [35]	245.53	63	6811	12.44	92.09	86.70	0.895	92.32
YOLOv4-tiny [42]	23.10	55	2802	40.58	93.11	31.83	0.418	67.43
MobileNetv1-YOLOv4	48.42	73	6486	18.77	93.39	57.87	0.665	88.61

**Table 3 sensors-22-06009-t003:** Comparison results of different model structures.

		Red Section			Green Section			Blue Section		
	Basic	SENet	ECANet	CBAM	SENet	ECANet	CBAM	SENet	ECANet	CBAM
Parameter size (MB)	48.42	49.07	48.42	51.04	48.49	48.42	48.70	48.58	48.42	49.07
FPS	18.77	18.46	18.86	16.20	17.49	17.42	16.21	17.64	18.67	16.65
Precision (%)	93.39	96.62	96.96	97.26	98.46	97.52	97.58	97.08	96.28	96.61
Recall (%)	57.87	65.27	70.53	58.36	73.97	70.70	74.86	64.38	67.92	66.51
*F*_1_ score	0.665	0.738	0.795	0.655	0.820	0.790	0.823	0.735	0.770	0.758
mAP (%)	88.61	93.03	92.57	87.93	93.85	92.72	93.76	91.93	90.42	89.18

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
