# Peer review of "Image Recognition of Wind Turbine Blade Defects Using Attention-Based MobileNetv1-YOLOv4 and Transfer Learning"

_sensors, 2022, doi:10.3390/s22166009_

Round 1
Reviewer 1 Report
Hi, thanks for your manuscript. I have the following concerns:
1. The literature review needs to be improved and compared with similar studies and what makes your study better?
Example:
Wang, Dong, Yanfeng Zhang, and Xiyun Yang. "Image Recognition of Wind Turbines Blade Surface Defects Based on Mask-RCNN." Advanced Intelligent Technologies for Industry. Springer, Singapore, 2022. 573-584.
2. The images shown in the paper are all from frontal view; however, in real scenarios using drones, the images can be taken at different angles ,distances, inclination,... which effects the accuracy to a very large extent. Have this been considered in the dataset? Please explain.
3. Could you please compare Tiny-yolov4 results with the proposed results in terms of accuracy and speed and add to table 2 ?
4. Could you please provide the qualitative results (visual results taken at real scenarios)?
5. Could you please add scenarios with individual sections (all three attention modules being included) removing other sections in Table 3?( e.g. including red section and removing green and blue sections )
6. Could you please explain why the parameter size of proposed method is far less than that of YOLOv3 (48 versus 245) but the FPS is lower (18.77 vs 19.38) considering the complexity of the attention-based module being negligible(mentioned by the authors)?
7. Based on the obtained results, the proposed method is not suitable and practical for real-time scenarios (18.77) which is an essential requirement for UAV-based use cases ? Could you please explain ?
Reviewer 2 Report
The proposed approach has novelty in methodology. Revision in terns of paper organization and technical details is needed. Consider following comments in the revised version.
1. Discuss about the runtime of your experiments briefly in the text. (new experiments are not needed)
2. Certainly add some images from dataset in the text. Some samples of defect and non-defect parts are needed.
3. I find the sentence "Therefore, 8-9 times less computation..." in the page 4. How do you select the number of less computation process epochs? discuss about it in a clear way.
4. Are the weights of less function parameters same in the Eq. 4? (class, confidence, IoU)
5. It is suggested to review related papers in visual defect detection scope. for example, I find a paper titled "Fabric defect detection based on completed local quartet patterns and majority decision algorithm", which has relation. Cite more classical or deep related works.
6. Is your proposed network accept a specific range of image size as input?
7. Your proposed deep network can be used in some other visual defect detection problems such as lytic bacteriophages detection in sewage water image. For example, I find a paper titled "Isolation and characterization of lytic bacteriophages infecting Pseudomonas aeruginosa from sewage water", which has enough relation. Cite this paper and discuss about this advantage of your proposed approach in the text.
8. I didn't find any over-fitting along with increasing training data in the Figure 14 . It is advantage. Discuss about it briefly in the conclusion
Round 2
Reviewer 1 Report
Hi , thanks for all the corrections, please include whatever you have added as your correction in the author respond to the manuscript. For instance , the images resulted from augmentation techniques( flipping, rotating,...) you have used and mentioned in the author response are missing and should be added to the manuscript . The edge detection you mentioned as the future work should also be added.
Author Response
Thanks very much for the comment from the reviewer. Considering the reviewer’s suggestion, we have added relevant corrections in the context. ‘An example of data augmentation for a crack defect is shown in Figure. 2 that 15 data enhancement methods are adopted.’ and ‘Moreover, the integration of attention-based modules with other detection algorithms also needs be explored further in the future, together with the edge detection method to directly give detection results by the UAV.’
Reviewer 2 Report
The authors' explanations are convincing. Sections have been added to the text that make the presented method easier to understand. The proposed method has enough novelty.
Author Response
Thanks very much for the approval from the reviewer. Thanks again for your professional review work on our article. Those comments are all valuable and very helpful for revising and improving our paper, as well as the important guiding significance to our researches.